# AI-IoT Low-Cost Pollution-Monitoring Sensor Network to Assist Citizens with Respiratory Problems

**DOI:** 10.3390/s23239585

**Published:** 2023-12-03

**Authors:** Santiago Felici-Castell, Jaume Segura-Garcia, Juan J. Perez-Solano, Rafael Fayos-Jordan, Antonio Soriano-Asensi, Jose M. Alcaraz-Calero

**Affiliations:** 1Escola Tècnica Superior d’Enginyeria, Universitat de València, Campus de Burjassot, 46100 Valencia, Spain; juan.j.perez@uv.es (J.J.P.-S.); antonio.soriano-asensi@uv.es (A.S.-A.); 2School of Computing, Engineering and Physical Sciences, University of West of Scotland, Storie Street, Paisley PA1 2HB, UK; rafael.fayos@uws.ac.uk (R.F.-J.); jose.alcaraz-calero@uws.ac.uk (J.M.A.-C.)

**Keywords:** air pollution, neural networks, low-cost sensors, IoT, WSN, forecasting, artificial intelligence, LSTM

## Abstract

The proliferation and great variety of low-cost air quality (AQ) sensors, combined with their flexibility and energy efficiency, gives an opportunity to integrate them into Wireless Sensor Networks (WSN). However, with these sensors, AQ monitoring poses a significant challenge, as the data collection and analysis process is complex and prone to errors. Although these sensors do not meet the performance requirements for reference regulatory-equivalent monitoring, they can provide informative measurements and more if we can adjust and add further processing to their raw measurements. Therefore, the integration of these sensors aims to facilitate real-time monitoring and achieve a higher spatial and temporal sampling density, particularly in urban areas, where there is a strong interest in providing AQ surveillance services since there is an increase in respiratory/allergic issues among the population. Leveraging a network of low-cost sensors, supported by 5G communications in combination with Artificial Intelligence (AI) techniques (using Convolutional and Deep Neural Networks (CNN and DNN)) to predict 24-h-ahead readings is the goal of this article in order to be able to provide early warnings to the populations of hazards areas. We have evaluated four different neural network architectures: Multi-Linear prediction (with a dense Multi-Linear Neural Network (NN)), Multi-Dense network prediction, Multi-Convolutional network prediction, and Multi-Long Short-Term Memory (LSTM) network prediction. To perform the training of the prediction of the readings, we have prepared a significant dataset that is analyzed and processed for training and testing, achieving an estimation error for most of the predicted parameters of around 7.2% on average, with the best option being the Multi-LSTM network in the forthcoming 24 h. It is worth mentioning that some pollutants achieved lower estimation errors, such as CO2 with 0.1%, PM10 with 2.4% (as well as PM2.5 and PM1.0), and NO2 with 6.7%.

## 1. Introduction

Citizens living in cities are constantly facing air pollution levels that are violating the human health safety thresholds defined by the World Health Organization (WHO) [1]. Based on recent studies issued by Eurostat [2], almost half a million EU residents die every year due to respiratory diseases in the group of 28 European countries. This reveals a huge number of deaths due to air quality (AQ) conditions in urban areas. Moreover, more than 90% of citizens are exposed to concentrations of toxic air pollutants above the WHO AQ Guideline (AQG) [3]. The WHO AQG defines the maximum levels of these pollutants to which humans should be exposed to avoid diseases and thus protect their health. Moreover, the problem is even worse when considering that the main part of the population has or may show respiratory and skin problems or allergies [4].

These environmental changes, due to urbanization and human impact, have effects on the atmosphere with three different aspects that are studied in different research fields: greenhouse gases, chlorofluorinated gases, and pollutants. Pollutants are the most critical since they are in contact with citizens, which include carbon monoxide (CO), ozone (O3), nitrogen dioxide (NO2), sulfur dioxide (SO2), as well as particulate matter (PM), with all of them in cities coming from the combustion of fossil fuels. The most harmful gases, according to the WHO, that citizens are exposed to in urban environments are PM, CO, O3, NO2, and SO2 [5]. Appendix A, lists these air pollutants, with a short explanation of their effects on human beings as well as the maximum levels proposed by AQG [3]. In Europe, we must stress that AQ is regulated by Directive 2008/50/EC and 2004/107/CE.

According to these directives, in particular, Directive 2008/50/EC on ambient AQ and cleaner air for Europe, the number of AQ monitoring points in each urban area should be at least one per 2 million inhabitants or one per 50,000 km^2^, where the latter criterion results in a higher number of monitoring points, but not less than one per area. To fulfill these rules, an official surveillance AQ monitoring network based on stations responsible for measuring polluting gases is deployed. In particular, in the Valencian Community (Spain), this network is operated by the Generalitat Valenciana [6] jointly with City Hall (for the stations deployed within cities). In particular, the official AQ stations in Valencia city are made available as open data through the Valencia minute-by-minute dashboard [7]. In Figure 1, some examples of these official AQ monitoring stations are shown, located in Burjassot and Valencia (Bulevar Sud) in the Valencian Community (Spain).

However, based on this information, the number of monitoring points is very small for estimating the pollution concentration on a street or in an area close to a citizen to assist them in their daily walks. This is the main reason that the deployment of low-cost AQ sensors is required. Moreover, this challenge is exacerbated when the system to address these problems needs to support city-scale wide-area coverage.

In this scenario, the utilization of 5G technologies, as shown in Figure 2 (with support to LoRa/Sigfox, WiFi, Bluetooth, Long-Term Evolution (LTE) for machines (LTE-M) and Narrow-Band IoT (NB-IoT)), along with Artificial Intelligence (AI) methods, in conjunction with the Internet of Things (IoT) have become crucial technologies to achieve these goals. Notice that in 2022, Sigfox went into bankruptcy proceedings, and Unabiz has taken control at the time of writing. Thus, by utilizing Wireless Sensor Networks (WSN) to monitor AQ with access to the Internet, under the requirements of low-cost deployments, a comprehensive system that focuses on the health concerns of citizens can be established.

Nevertheless, addressing AQ poses a significant challenge as the process of measuring, analyzing data across various locations is complex and prone to errors [8]. Therefore, in this article, we present enhancements made to traditional monitoring systems by taking advantage of existing infrastructures and optimizing them with our bespoke collection network. This integration incorporates low-cost components to significantly improve the spatial sampling density, processing raw data with AI techniques and providing a more comprehensive and detailed view of AQ. The AI-driven proposed framework allows the prediction of the values of the pollutants up to 24 h ahead to provide the population with early warning of potential exposures to pollutants.

The rest of the paper is structured as follows. Section 2 introduces a thorough revision of the state-of-the-art and related work regarding AQ sensors, solutions, and similar results or research projects. Section 3 shows the architecture of the proposed system for collecting measurements from low-cost sensors. Section 4 focuses on the explanation of the use cases envisaged for the AI-IoT system. Section 5 describes the process of predicting raw measurements supported by AI algorithms to improve the performance of the IoT system. Finally, Section 6 concludes the paper and describes future work.

## 2. State of the Art

As seen in the previous section, increased awareness of AQ due to its health effects has led to a boom in low-cost sensors to increase AQ monitoring density. Due to their ease of installation and low power consumption, they are interesting in terms of integration into WSN.

Since there are many types and models of low-cost AQ sensors, it is difficult to review all of them in detail. Moreover, this market is highly dynamic. These low-cost AQ sensors can measure pollutants, such as the ones mentioned before, along with temperature (T), atmospheric pressure (AP), and relative humidity (RH). Figure 3 shows examples of these low-cost sensors; from left to right, (MiCS5524) [9], PM (Plantower PM2.5) [10], and/or low-cost CO2 sensors (MHZ19B) [11], that can measure PM (PM2.5, PM10) and gases (O3, total nitrogen oxides (NOx) (nitrogen monoxide (NO), nitrogen dioxide (NO2)), SO2, CO2, and CH4.

Also, these low-cost sensors are embedded in ad hoc modules, such as the ones shown in [12,13], known as Real-time Affordable Multi-Pollutant (RAMP) sensor packages, that can measure CO, NO2, O3, and CO2 components, making them easier to use and therefore more attractive. Other commercial alternatives to these modules and their main characteristics, as well as the type of data connection, are shown in Table 1. From all of these options, we have chosen the module ZPHS01B [14] because it has an attractive set of already embedded sensors, is ready to use, and provides the largest number of gases (with support to detect CO2, CO, CH2O, NO2, O3 and TVOC sensors and PM) for AQ monitoring, as well as the best quality/price ratio. Figure 4 shows a picture of this ZPHS01B AQ module sensor.

It is worth mentioning that, depending on their operating principle, these sensors are available in different technologies to react to the presence of different pollutants: electrochemicals, metal oxide semiconductors, photoionization detectors, non-dispersive infrared, and light scattering, among others. A review of the performance of these low-cost sensors can be found in [8]. In this study, the Pearson Correlation Coefficient is used to describe how well the response of these sensors correlates to the reference instruments.

Although these low-cost sensors do not meet the performance requirements in terms of accuracy and electromagnetic compliance for reference regulatory-equivalent monitoring, they can provide informative measurements and more if we can adjust and add further processing to their raw measurements. However, a notable development in this context is the release of the standard CEN/TS 17660-1:2021 [18], which outlines the criteria specified by Directive 2008/50/EC for assessing the equivalence of sensor systems employed in outdoor settings to conventional instruments used for indicative measurements and objective estimation [8].

In practice, these sensors offer the capability to provide a rough estimate or general understanding of AQ and enable the identification of areas with high pollution levels. However, to improve the accuracy of the readings, the measurements obtained from these sensors can be incorporated into the modeling process alongside other data, such as measurements of additional pollutants and ambient conditions, such as T and RH. This integration of data sources helps to improve the reliability of the overall assessment, as we will see in Section 5 using AI techniques to improve the performance of the IoT system by applying forecasting techniques for their estimation.

It is worth mentioning that to use AI techniques, we should consider and follow the recommendations given in [19]. In this reference, the best practices and common pitfalls in the use of machine-learning techniques for environmental research are suggested. It is suggested that it is good practice to compare at least two supervised learning methods to justify the method selection. Also, the process of creating the dataset and data-splitting into three different sets for training, validation, and testing is defined. Moreover, other issues are discussed, such as the proper sample size and feature size, data enrichment and feature selection, randomness assessment, data leakage management, method selection and comparison, model optimization and evaluation, and model explainability and causality as well as benchmarking metrics related to the error estimation techniques, based on Mean Absolute Error (MAE), Root Mean Square Error (RMSE) and $R^{2}$. Thus, good practice involves using more than one metric to comprehensively assess the model performance.

In particular, AQ forecasting for environmental pollution monitoring with low-cost AQ sensors can be applied in combination with AI techniques. A number of publications have envisaged this issue, considering different points of view. In [20], the authors made a bibliometric literature review of the applications for air quality forecasting with AI. They conclude that although the applications are growing, the number of publications is still limited. In [21], the authors did a narrative review of the state-of-the-art in AQ metrics forecasting. In [22], the authors used Long Short-Term Memory (LSTM) and Convolutional Neural Networks (CNN) approaches for AQ metrics prediction. In [23], the authors developed an AQ IoT system with AI predicting features, but they are more focused on machine-learning techniques rather than on CNN or Deep Neural Networks (DNN).

Regarding AQ applications, based on these low-cost sensors, we can see interesting use cases as shown in [24]. This application is oriented to measure ventilation quality by measuring CO2 levels and T/RH values. Also, applications to estimate and analyze healthy routes for pedestrians and bikers, based on the AQ information gathered by the proposed AQ monitoring network, are shown in [25,26].

Finally, we can mention commercial initiatives, such as [27,28], based on a similar approach, although with a purpose limited only to the monitoring of pollutant gases, without attempting to cover and analyze the problem posed at the urban level as a whole holistically.

## 3. Design Alternatives and Techniques to Be Used in the AQ Monitoring Network and Its Architecture

The proposed monitoring network is composed of wireless AQ sensor nodes based on a microcontroller connected to the AQ module ZPHS01B [14], enabling the use of any of the communication technologies depicted in Figure 2. These nodes also incorporate real-time clocks, external memory cards, and watchdog mechanisms.

In particular, the ESP32 microcontroller has been selected due to its performance and good quality/price ratio. It must be stressed that based on this microcontroller, we have commercial modules, such as FiPy module [29,30] by Pycom Ltd, which includes onboard technologies such as LTE-M/NB-IoT, LoRa/Sigfox, WiFi, and Bluetooth.Notice that Pycom Ltd. went into administration in September 2022, but the newly created Pycom BV has taken over this company to prevent the products from going to end of life at the time of writing.

In Figure 5, we show a picture of this microcontroller with the detail of the connection to the ZPHS01B sensor module, and in Figure 6, the proposals of our low-cost indoor and outdoor prototypes are shown, with their cases for AQ monitoring. Both prototypes use the same circuit board and sensor module, with a small fan at the top of the plastic tube and an air inlet at the bottom of the tube that draws in air. This fan creates an airflow, according to the specifications [14], to create a gentle air pressure on the sensors. Moreover, to protect the whole system outdoors, we keep the head of the system with a lid or cover, as shown in Figure 6b, allowing air circulation. The microcontroller is within a sealed plastic box.

Figure 7 shows the system architecture of the whole system. The communication scheme between the IoT node and the infrastructure relies on the IoT Message Queue Telemetry Transport (MQTT) protocol, which enables the transmission of information through messages exchanged between the nodes and the MQTT broker. It is important to emphasize that MQTT offers three levels of Quality of Service (QoS) to ensure message delivery and incorporates various security mechanisms for data transmission. In our implementation, we have opted for the highest QoS level, QoS-2, which guarantees the delivery of messages. Also, we employ username and password-based authentication for both the broker and the clients, using SSL-certified encryption to safeguard the transmitted data. The received data are locally stored in a database. To facilitate the publishing process, nodes can create new topics simply by publishing them. This feature allows for the easy integration of additional nodes into the IoT system, greatly enhancing its scalability. This MQTT-based architecture allows direct integration with LTE-M. NB-IoT and WiFi. For Bluetooth and LoRa, an inter-medium gateway is being used to receive these messages and relay them to an IP-based network using LTE-M, NB-IoT, or WiFi.

On the server side, we run an IoT platform. Within the IoT platform, we use an ingestion server to manage the metrics transmitted through the communication protocol used, MQTT, and to store this information in the InfluxDB database in order to be able to operate with the metrics. For this purpose, we use Telegraf [31], in charge of receiving the metrics and collecting them into the DB. The InfluxDB database is an open-source no-SQL time-series database that allows the management of information in a more agile and accessible way since the way of operating does not consist of the classic primary key in tables, but the indexing is conducted by means of a tag and a timestamp [32]. For the visualization part, we use Grafana [33], an open-source multiplatform web application for the analysis of metrics that allows the creation of graphs from various sources in an easy, convenient, and simple way.

It should be noted, as mentioned before, that these direct measurements (raw readings) taken from the sensors, in order to be considered valid, are also inserted in the training server to perform the training of an AI model used for the prediction for the next day (24 h). Such a model is being used in the inference server to perform the forecasting of such metrics values, and it is being updated at constant interval rates.

For management purposes, Figure 8 shows a screenshot of the interface connection via Bluetooth to check locally and directly the status of each IoT node. This information can also be accessed remotely. The information shown is sent periodically every 10 min. It includes ten samples (raw readings) from all AQ sensors embedded in module ZPHS01B with the measurements from the different sensors [14]. In this Figure 8, the raw readings from the low-cost module ZPHS01B AQ sensor board with 11 different sensors are shown. These readings are PM (1.0, 2.5, 10), CO2, TVOC, T, RH, CH2O, CO, O3, NO2, and hours and minutes in this order. In ug/m3, PM (1.0, 2.5, 10) and CH2O are given. CO2, CO, O3, and NO2 are given in parts per million (ppm). T and RH are given in degrees (Celsius) and %, respectively.

From a practical point of view for the deployment, when we connect these sensors off the shelves, we place them next to the AQ official monitoring station. These low-cost sensors need calibration. We perform the calibration process for each one of the different sensors using these official measurements. Notice that we only focus on the usual values as the ones we obtain from these stations, i.e., we do not perform a complete whole range calibration in the lab using an air quality calibration camera with the different gases. The calibration process is checked and adjusted every day to prevent variances. It is worth mentioning that the lifetime of these sensors is short, usually between 12 and 15 months, depending on the exposure and conservation. In practice, we periodically replace these modules with new ones when we see that the calibration process does not work. This usually happens after approximately one year of use. The power consumption for these low-cost AQ monitoring nodes is approximately 200 mA with 5 V (1 W). In this case, we use a direct power supply from the station.

We use 24-h forecasts for several reasons, although a similar approach could be done for 1 and 6 h. On the one hand, these low-cost nodes require periodic calibrations every 24 h to maintain the accuracy of the measurements. On the other hand, we must stress that in 24 h, we can monitor the complete cycle of pollutants as well as the complete cycle of citizen movements, which are highly correlated with the pollutant emissions that allow us to model the forecast prediction properly.

## 4. Use Cases

The AQ application designed to assist citizens is shown in [25,26]. This application analyzes, evaluates, and processes the air pollution estimation along a specified route. When the user (citizen) requests a specific route to this application, from a source to a destination, initially, we check his/her user profile (medical history) to extract information about the risks given to this citizen for each air pollutant. These risks, for this user, are given in percentages according to their danger to him/her. These percentages are specified for the different pollutants that determine a weighted value associated with the total risk. These total values are mapped over a grid using interpolation techniques that are later placed into Open Street Maps to determine a complex metric to search for the route with less pollution or to analyze the pollution along a given route.

Thus, our purpose for the use of the AQ IoT monitoring system in combination with the AQ application is envisaged in two use cases that have been previously studied. These are:1.Early-warning system and location: this use case was studied in [25], where we established a route and analyzed the statistics of exposure to different air pollution metrics. The use of an AQ forecasting module will help detect specific hot areas for air pollutants in the following hours and send warnings to citizens.2.Pollutant-aware route planning: this use case was studied in [26], where we applied the AQ monitoring and spatial statistics to the evaluation of the best routes for air pollution minimization. Also, the use of an AQ forecasting module will help schedule a route previously and select the best route option to minimize exposure.

## 5. Use of Neural Networks to Forecast Air Pollution Data Metrics

This section describes the process for enhancing and predicting valid AQ measurements from raw sensor readings supported by AI algorithms. Notice that as stated in Section 1, for this process, we use the information coming from two different data sources: our proposed AQ collection network and the official AQ measurements. These two data sources are used to train Neural Networks (NNs)—in particular, CNN and DNN. Once the CNN or DNN is trained, parameter forecasting is achieved and enabled in the proposed architecture.

The selection of the appropriate NNs will help to ensure the reliability of the low-cost sensor data, which remains one of the main challenges, as mentioned above. To address this issue, it is crucial to establish data consistency between sensor measurements (raw readings) and those obtained through standard measurements, a process commonly known as Quality Assurance (QA). For this process, we must place the proposed AQ nodes next to the official AQ monitoring stations. Figure 9 shows two examples of the nodes deployed to collect raw data and compare it with these stations. As we can see in this figure, our prototypes are placed on the top and next to the official AQ station. Calibration techniques are frequently used to compare sensor readings with those of reference-grade instruments, as explained in Section 3.

In this context, NNs have proven to be very useful in enhancing measurements from low-cost sensors and forecasting values to enable proactive actions over pollutant exposure, as explained in Section 2. This is due to the ability of CNN and DNN (in particular, we use LSTM networks [34]) to learn behaviors that appeared during the training process to forecast them. We can see a NN as a combination of different layers of neurons. With adequate training, this combination of layers learns a mapping from inputs to outputs according to the training set. These layers individually provide us with an output (i.e., the forecasted metric values) as a function of an input (i.e., the measured metrics), and since they are interconnected, we can obtain a given output as a function of a given input to the NN.

CNNs are based on the utilization of convolutional layers, which execute various filtering processes. The filters applied at each layer are determined through the training stage by adjusting the network weights that try to minimize a predefined loss function. It generates internal representations based on a set of optimized network weights, representing learned filters focused on a certain issue. The raw readings (raw samples) from low-cost sensors serve as the input to the model. Thus, this network analyzes and extracts relevant features across successive stages, resulting in relevant representations from these learned filters. The filters within the convolutional layers have a length given by the local receptive field of a single unit within that layer. The application of these filters to the input samples produces the layer’s output, also called feature maps. The key aspects of CNNs include parameter sharing (weights shared by all neurons on a specific feature map) and local connectivity (each neuron is connected to only a subset of input nodes). The reduction of the model parameters contributes to maintaining the same feature detector in different sections of the input data.

DNNs, and in particular LSTM networks [34], are a type of Recurrent Neural Network (RNN) architecture designed to address the vanishing gradient problem in traditional RNNs. LSTM networks are equipped with a more complex and efficient gating mechanism that allows them to capture and remember long-term dependencies in sequential data. This gating system, which includes the input, forget, and output gates, controls the flow of information through the network, enabling it to update and retain information over extended sequences selectively. LSTM networks are particularly suited for tasks involving time-series data, natural language processing, and any problem where the context from previous time steps is critical for making accurate predictions or classifications, as our proposed AQ predicts and improves the process.

It is worth mentioning that we address the design for parameter predictions of the CNN and DNN as a regression problem. Thus, the objective is to obtain an accurate end-to-end architecture capable of predicting the whole set of parameters from raw AQ inputs.

### 5.1. Dataset

To carry out the training of the NN, it is necessary to have a large database of different metrics involved in the AQ measurements. For this, we use the available variables given by the proposed AQ monitoring nodes, as well as the official AQ monitoring stations (operated by the Generalitat Valenciana [6] and the City Hall of Valencia [7]), such as T (${}^{{}^{\circ}}$C), RH (%), PM (PM1.0, PM2.5 and PM10.0), VOCs, Formaldehyde (CH2O), CO2, O3 and NO2, with a lot of timely variations, since the parameters we want to enhance depend on many external and environmental factors. As mentioned before, they have been measured in two specific locations within the station, as depicted in Figure 9, as close as possible to those at which the measurements are done. These measurements were made between 26 June 2023 and 3 September 2023, with 10 sensor readings every second. Notice that the CO sensor was not considered as it was not functional in this case. Thus, we only work and forecast with 10 different sensors, namely PM(1.0, 2.5, 10), CO2, TVOC, T, RH, CH2O, O3, and NO2. Then, we obtained 100,7540 samples of each one of the previous metrics for the whole period. Figure 10 shows the measurement record of the different metrics used. ug/m^3^ are given PM(1.0, 2.5, 10). and CH2O. CO2, CO, O3, and NO2 are given in PPM. T and RH are given in degrees Celsius (${}^{{}^{\circ}}$C) and %. In addition, they have been averaged hourly (every 600 samples), obtaining 1680 samples. Figure 11 shows the violin diagram of the normalized different readings used to train the NNs. Table 2 depicts the mean, standard deviation, min, and max values of the raw readings from these sensors.

To facilitate the training and evaluation of the NN, the complete dataset was divided into three distinct partitions: training, validation, and testing. The training and validation datasets were exclusively used during the training phase, consisting of 70% of samples for training and 20% for validation. Additionally, a separate test partition comprising 5000 samples was included to conduct a more comprehensive assessment of CNN performance. It is important to note that the samples within this final test partition were not involved in the training or validation processes.

### 5.2. Design, Configuration, and Training

A good practice to analyze the different AI models is to compare at least two supervised learning or statistical methods to justify the method selection. In our case, we used four [19]. Thus, the evaluation of the four different options for our dataset will help us to select our best option for environmental pollution forecasting. In this case, we use Multi-Linear prediction (with a dense Multi-Linear NN), Multi-Dense network prediction, Multi-Convolutional network prediction, and Multi-LSTM network prediction, denoted as Linear, Dense, Conv., and LSTM, respectively. All hyperparameters associated with each of these AI models are depicted in Table 3, Table 4, Table 5 and Table 6, in the same order given before, with details of the layer type, output shape, number of parameters, and size (KB).

The training process was limited to a maximum of 20 epochs with a mechanism of Early Stopping (ES), which monitors the “validation loss” to minimize it and escape from this training process before reaching Epoch 20. MAE was chosen as the loss function to be minimized. It was computed by calculating the average value of the differences between the predicted and actual values. We must stress that in both the loss model for the training and validation process, we see a convergence to a minimum MAE by increasing the number of epochs. However, in the validation process, we notice an added variability to this convergence process due to the quality of these AQ sensors.

To prevent overfitting, an ES technique was employed, which saved the best-performing model based on its performance on the validation partition. The ES event took place at Epoch 10. Figure 12 shows the MAE of the test and validation for multimetric training. In this figure, we can see that the best option to do the multimetric environmental pollution forecasting is provided by the LSTM network, with an MAE equal to 0.5166 with the test dataset, which is lower than the other ones.

The subsequent results obtained from testing the model on the separate test partition, as well as its performance during a real deployment of the AI-IoT network, are shown in Figure 13, Figure 14, Figure 15, Figure 16 and Figure 17. In particular, these figures are a sample to show the T, RH, PM.25, CH20, and CO2 forecasting, respectively, for 24 h with the trained LSTM model. At each figure, we plot three different time slots as examples, with time offsets of 100 h between them.

The summary of the evaluation in the training, validation, and inference process, as well as the size of the weights obtained for each network, is depicted in Table 7. From this table, we can see that LSTM is the most cost-effective network for use in AI-IoT devices, followed by Convolutional, Multi-Linear, and Multi-Dense. For instance, for the T prediction with the LSTM model, we obtain an MAE of 0.5166, which is a 2.0% relative error with a mean T of 25.65${}^{{}^{\circ}}$. In detail, for each parameter, we can see in Table 8, the MAE achieved, that on average, shows an estimation error of around 7.2%. Notice that the achievement of an AI model that fits both in an inference server and IoT devices indistinctly is another achievement of our contribution, and it allows the future exploration of dynamic computational movements between the inference server and the IoT devices. Moreover, Table 9 adds more quantitative performance metrics to strengthen the results, including MAE, MSE, and RMSE, in order to better characterize the model accuracy. As we can see, the behavior of the LSTM network is better in terms of MAE, MSE, and RMSE compared with the other alternatives.

Furthermore, as we can see in Table 7 if we compare the model size, the LSTM model with 189 KB has a larger size than Multi-Linear (52 KB) but lower than Convolutional (842 KB) and Multi-Dense (1535 KB). For the inference time, all of them have a similar time response. However, for the training and validation process, the LSTM model is the most time-consuming, 7.34 s compared with 2.61, 2.91, and 4.40 s for Convolutional, Multi-Dense, and Multi-Linear, respectively. They all provide very acceptable MAE levels, showing interesting alternatives to be used as a trust estimator of the potential forecasting of the values related to the pollutants, but the LSTM model gives the lowest MAE. Thus, we can consider it to be the most accurate network for our purpose.

## 6. Conclusions and Future Work

Generally, low-cost AQ sensors do not meet regulatory requirements for equivalent monitoring. Their sensitivity, time response, and accuracy are very limited, but through data augmentation with official open data coming from official AQ monitoring stations and AI techniques, we can improve their measurements to increase the accuracy of pollutant distribution and forecast these environmental pollution metrics to achieve early-warning systems for hazard pollutant exposure of the population.

The use of DNN techniques, in particular LSTM networks, has been proven to be effective in forecasting pollution metrics with an averaged estimation error of 7.2%. Notice that the behavior of the LSTM network is better in terms of MAE, MSE, and RMSE compared with the other alternatives. It is worth mentioning that some pollutants have lower estimation error, such as CO2 with 0.1%, PM10 with 2.4% (as well as PM2.5 and PM1.0), and NO2 with 6.7%, as shown in Table 8. However, the AQ monitoring performance for some pollutants registered by ZPHS01B [14] was not good, which was the case for VOC and CH20 components.

Thus, we must stress that with a deployed WSN for AQ monitoring, like the one proposed in [26] and its collected information, we can inform citizens of potential exposure to dangerous pollution levels before they are even being exposed, as well as to help and assist them to plan pollutant-aware routes in order to minimize their impact and exposure to pollution.

Notice that the achievement of an AI model that fits both in an inference server and IoT devices indistinctly is another achievement of our contribution, and it allows the future exploration of dynamic computational movements between the inference server and the IoT devices.

As future work, we are currently preparing enhanced datasets and analyzing their behavior so that we can extrapolate improved metrics using AI techniques for the entire sensing network. In addition, we are considering using better AQ monitoring modules. Finally, it must be noticed that there are other alternative and complementary metrics, such as subjective noise annoyance, which can be part of these route planning algorithms too [35].

## Figures and Tables

**Figure 1 sensors-23-09585-f001:**
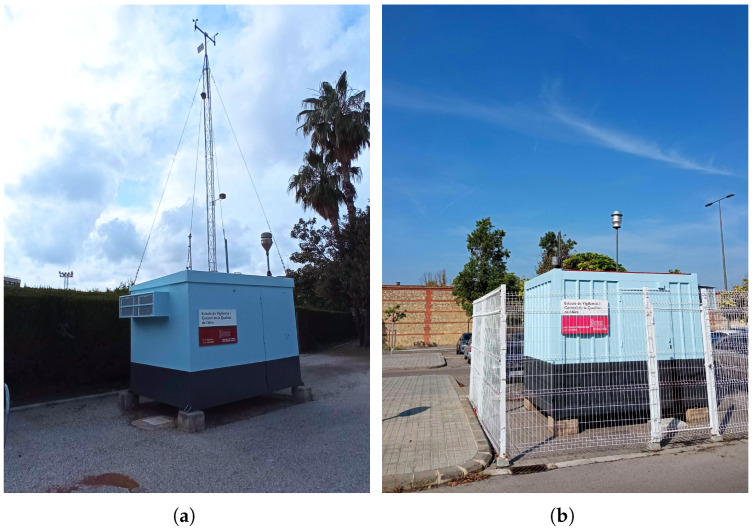
Example of official AQ monitoring stations: (**a**): Burjassot city (Spain) and (**b**): Valencia city, Bulevar Sud (Spain).

**Figure 2 sensors-23-09585-f002:**
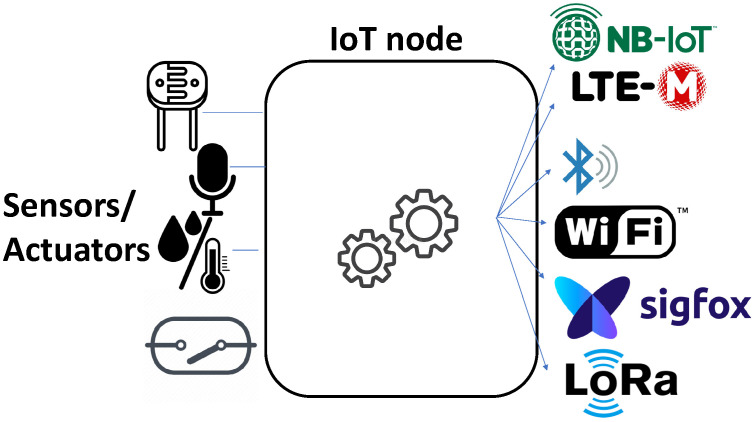
Proposal of a generic IoT node for AQ monitoring and its communications support.

**Figure 3 sensors-23-09585-f003:**
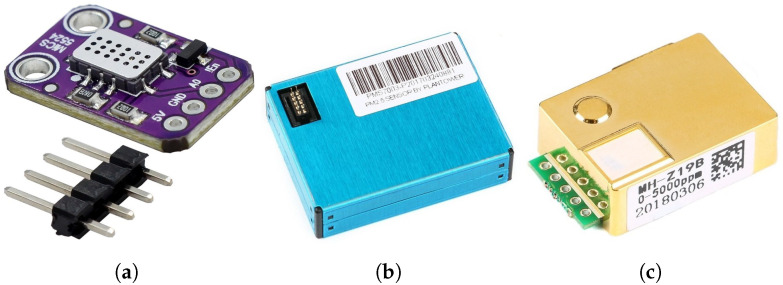
Examples of low-cost sensors. From left to right, air quality (MiCS5524) [9], PM (Plantower PM2.5) [10], and/or low-cost CO2 sensor (MHZ19B) [11]. (**a**) AQ (MiCS5524); (**b**) PM2.5 (Plantower); (**c**) CO2 (MHZ19B).

**Figure 4 sensors-23-09585-f004:**
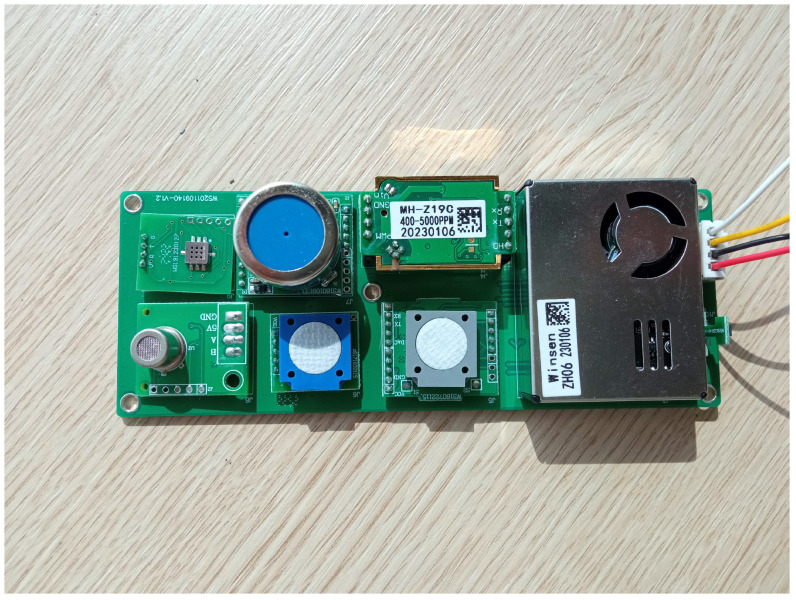
Photo of the low-cost module ZPHS01B AQ sensor board [14] with support to detect PM, CO, NO2, SO2, O3 and Total Volatile Organic Compounds (TVOCs) sensors.

**Figure 5 sensors-23-09585-f005:**
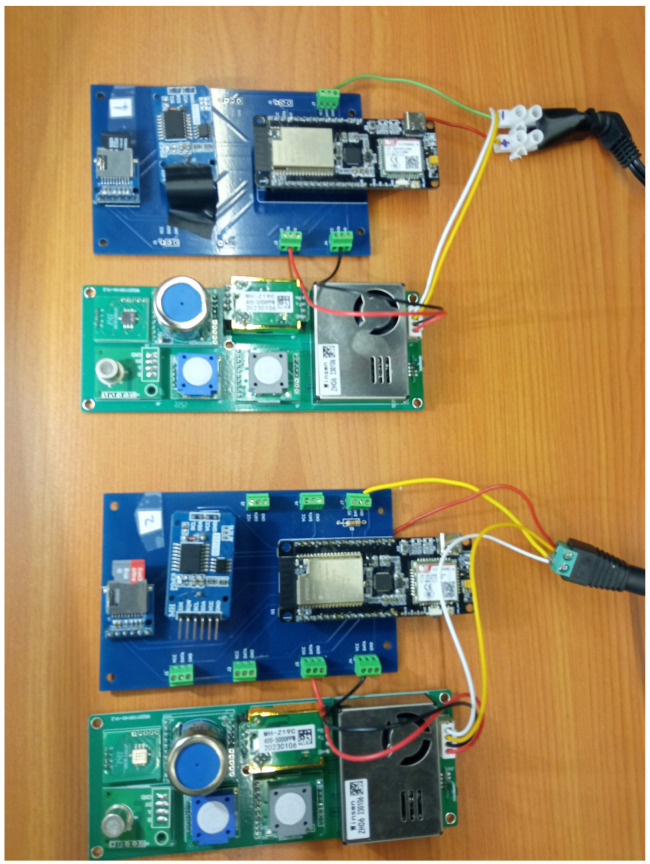
Example of two bare AQ monitoring nodes with ESP32 microcontroller connected to the AQ module ZPHS01B [14] sensor board.

**Figure 6 sensors-23-09585-f006:**
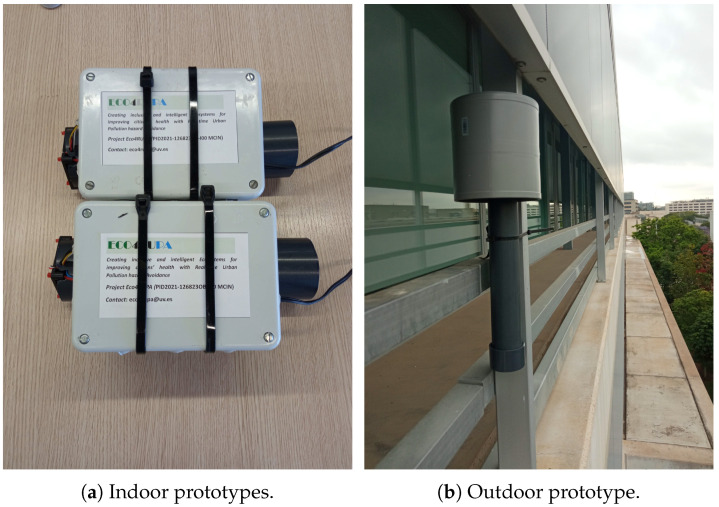
Low-cost AQ monitoring indoor and outdoor prototypes.

**Figure 7 sensors-23-09585-f007:**
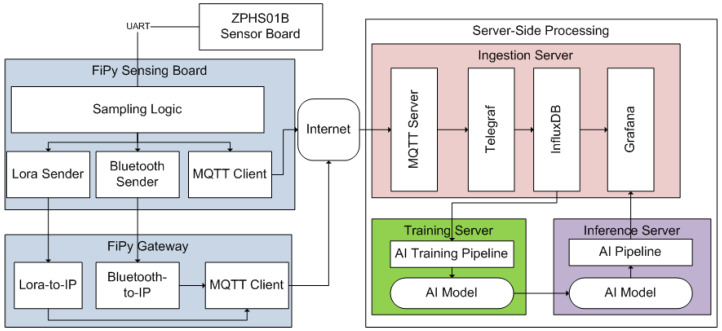
Overview of the system architecture and the IoT platform.

**Figure 8 sensors-23-09585-f008:**
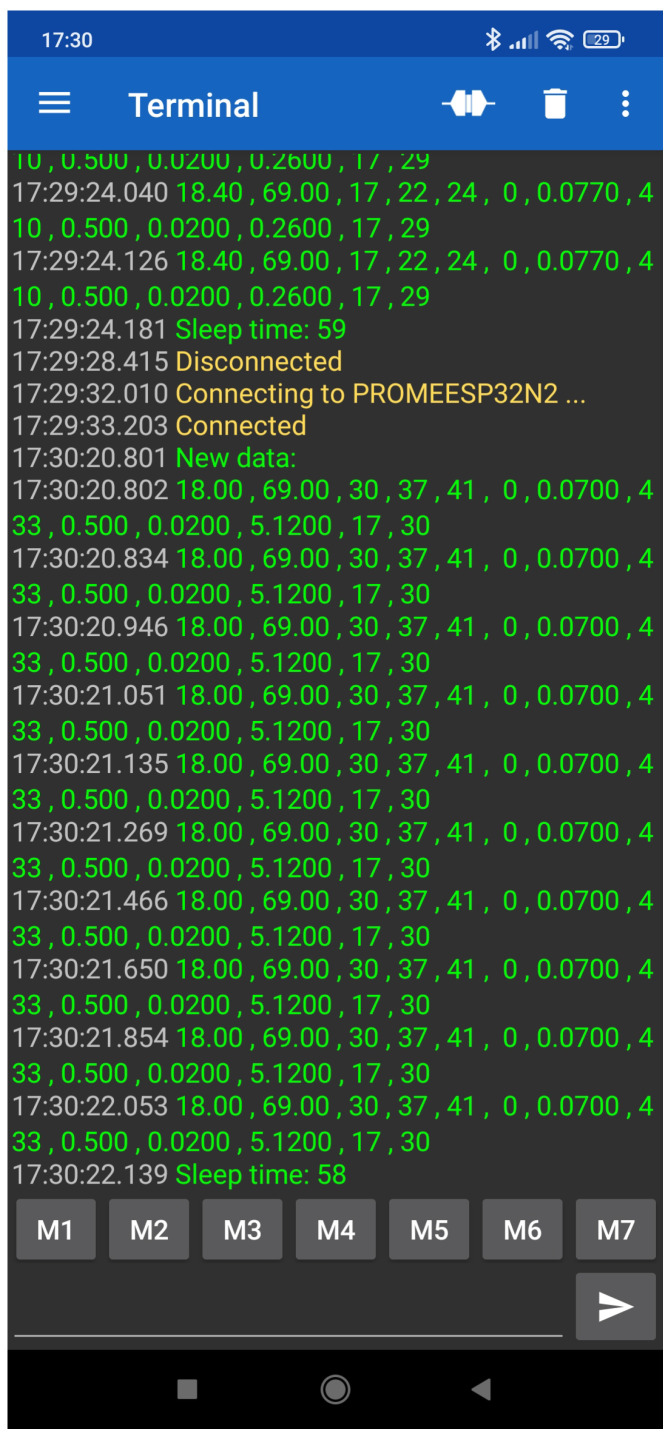
Screenshot of the management interface to access the IoT AQ nodes.

**Figure 9 sensors-23-09585-f009:**
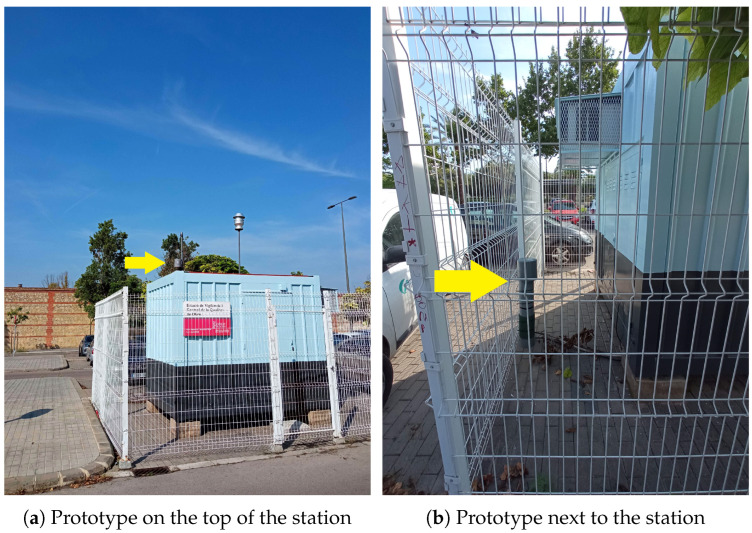
Deployment of low-cost AQ monitoring nodes for calibration and training in the official AQ station at Bulevar Sud (Valencia), (**a**): on the top and (**b**): next to. The yellow arrow indicates the exact location of the low-cost node.

**Figure 10 sensors-23-09585-f010:**
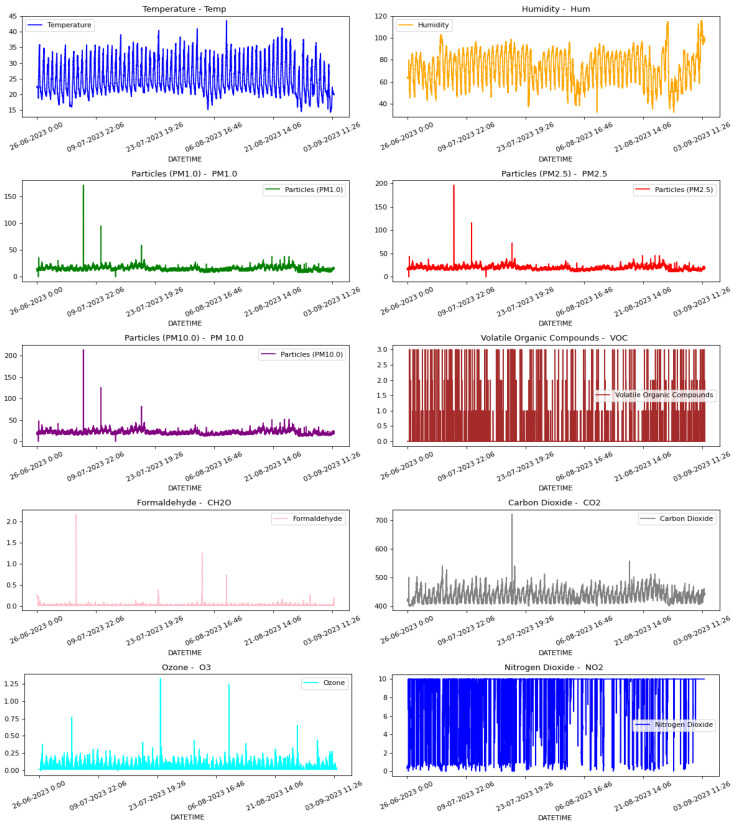
Measurement record of the different metrics used to train our DNN based on an LSTM network.

**Figure 11 sensors-23-09585-f011:**
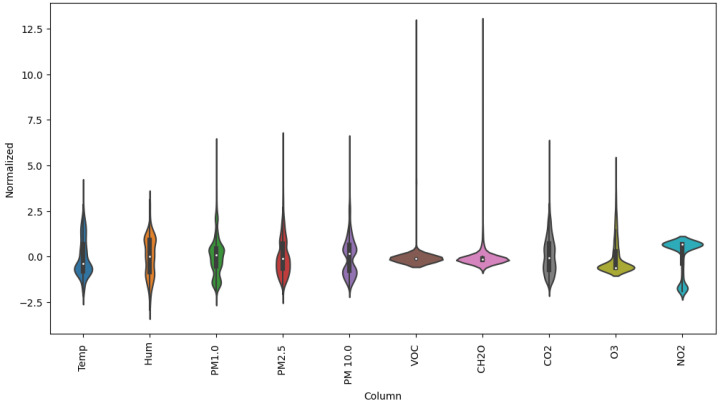
Violin diagram showing the statistics of the different metrics used to train the NNs.

**Figure 12 sensors-23-09585-f012:**
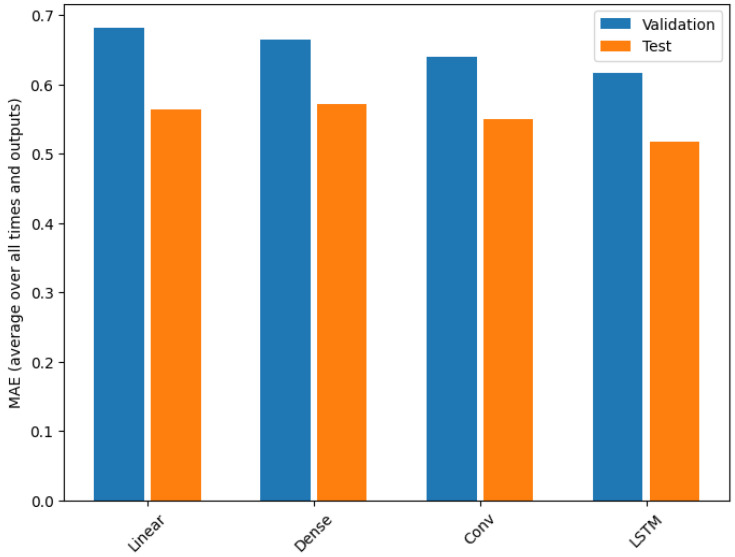
MAE with different NN (such as Linear, Dense, Conv. and LSTM) for AQ metrics forecasting using the validation and test datasets.

**Figure 13 sensors-23-09585-f013:**
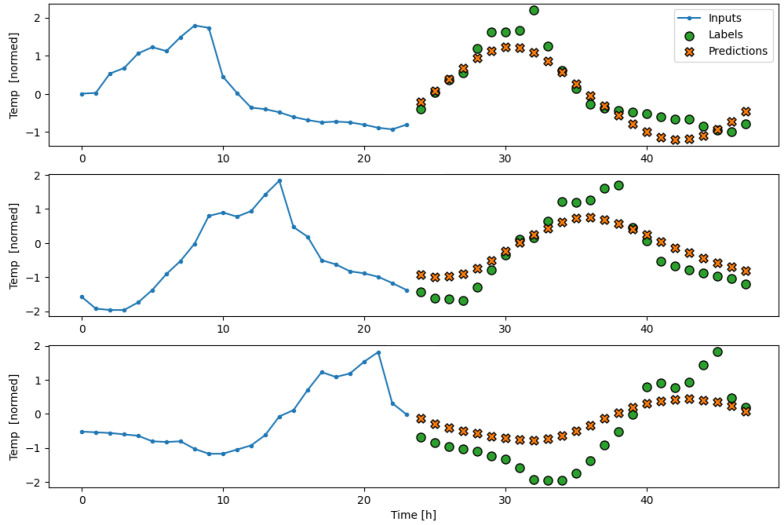
Example of T forecasting for 24 h with the trained LSTM model.

**Figure 14 sensors-23-09585-f014:**
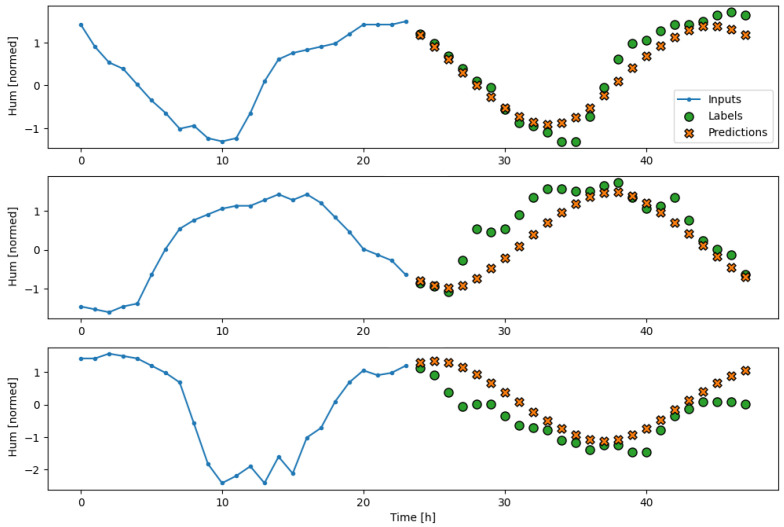
Example of RH forecasting for 24 h with the trained LSTM model.

**Figure 15 sensors-23-09585-f015:**
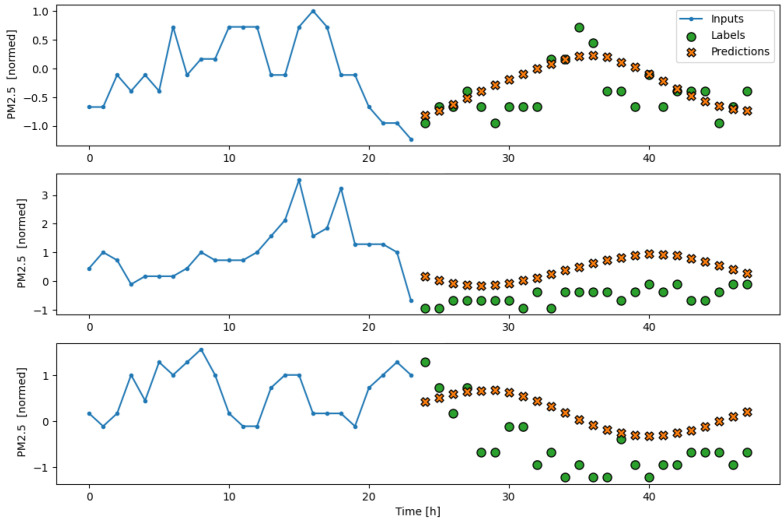
Example of PM2.5 forecasting for 24 h with the trained LSTM model.

**Figure 16 sensors-23-09585-f016:**
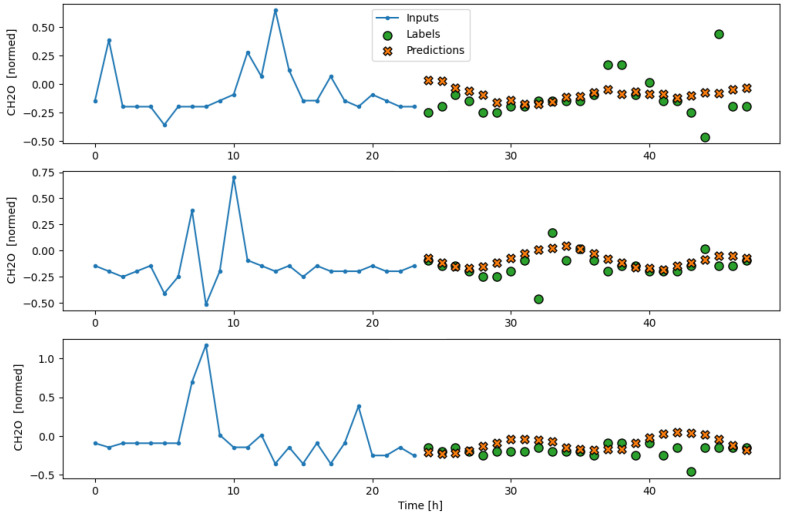
Example of CH2O forecasting for 24 h with the trained LSTM model.

**Figure 17 sensors-23-09585-f017:**
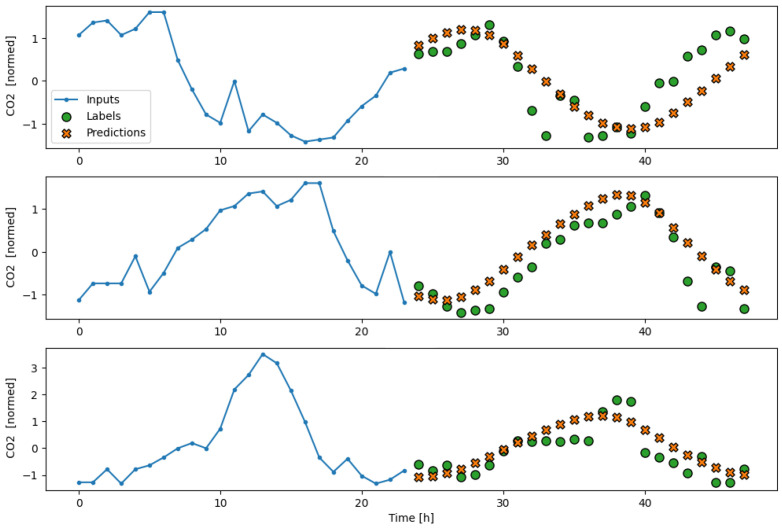
Example of CO2 forecasting for 24 h with the trained LSTM model.

**Table 1 sensors-23-09585-t001:** Sensor comparison for AQ.

Module	Detected Gases and PM	Connectivity
SDS011 [15]	PM, T, HR, PA	UART
DL-LP8P [16]	CO2, T, HR, PA	LoRAWAN
MiCS-6814 [17]	CO, NO2, C2H5OH, NH3, CH4	I2C, SPI
ZPHS01B [14]	PM, CO2, CO, CH2O, O3, NO2, TVOC, T, HR	UART

**Table 2 sensors-23-09585-t002:** Statistics of the different metrics used to train our DNN, based on an LSTM network. In ug/m^3^ are given PM(1.0, 2.5, 10) and CH2O. CO2, CO, O3, and NO2 are given in ppm. T and RH are given in ${}^{{}^{\circ}}$ and %.

	Mean	Std	Min	Max
Temp (${}^{{}^{\circ}}$)	24.86	5.17	14.5	43.6
Hum (%)	72.67	14.63	33	115
PM1.0	15.68	3.27	9	34
PM2.5	19.43	3.81	12	42
PM10.0	21.29	4.41	14	47
TVOC	0.03	0.24	0	3
CH2O	0.02	0.02	0.01	0.26
CO2	436.57	20.63	400	557
O3	0.04	0.03	0.02	0.21
NO2	7.72	3.73	0.01	10

**Table 3 sensors-23-09585-t003:** Multi-Linear network definition (Linear).

Layer (Type)	Output Shape	Param #
lambda (Lambda)	(None, 1, 10)	0
dense (Dense)	(None, 1, 240)	2640
reshape (Reshape)	(None, 24, 10)	0
Total params:	2640 (10.31 KB)	

**Table 4 sensors-23-09585-t004:** Multi-Dense layer network definition (Dense).

Layer (Type)	Output Shape	Param #
lambda (Lambda)	(None, 1, 10)	0
dense (Dense)	(None, 1, 512)	5632
dense (Dense)	(None, 1, 240)	123120
reshape (Reshape)	(None, 24, 10)	0
Total params:	128,752 (502.94 KB)	

**Table 5 sensors-23-09585-t005:** Convolutional network layers definition (Conv.).

Layer (Type)	Output Shape	Param #
lambda (Lambda)	(None, 3, 10)	0
conv1d (Conv1D)	(None, 1, 256)	7936
dense (Dense)	(None, 1, 240)	61680
reshape (Reshape)	(None, 24, 10)	0
Total params:	69,616 (271.94 KB)	

**Table 6 sensors-23-09585-t006:** LSTM network layers definition.

Layer (Type)	Output Shape	Param #
lstm (LSTM)	(None, 32)	5504
dense (Dense)	(None, 240)	7920
reshape (Reshape)	(None, 24, 10)	0
Total params:	13,424 (52.44 KB)	

**Table 7 sensors-23-09585-t007:** Summary of the performance in the Training/Validation (T/V) and inference process with the different NN (Linear, Dense, Conv. and LSTM) for T forecasting.

	Multi-Linear	Multi-Dense	Convolutional	LSTM
MAE test	0.5643	0.5742	0.5458	0.5166
Model size (KB)	52	1535	842	189
T/V time (s)	4.40	2.91	2.61	7.34
Inference time (s)	0.11	0.12	0.13	0.12

**Table 8 sensors-23-09585-t008:** Summary of MAE evaluation for every metric in the test phase with the different NN models.

	Temp	Hum	PM1.0	PM2.5	PM10.0	TVOC	CH2O	CO2	O3	NO2
MAE/mean norm	2.0%	0.7%	3.3%	2.7%	2.4%	17.3%	26.0%	0.1%	12.8%	6.7%
MAE Linear	0.5638	0.564	0.5638	0.564	0.5643	0.5642	0.5633	0.5647	0.5649	0.5645
MAE Dense	0.5719	0.5544	0.5751	0.5689	0.5654	0.5659	0.5567	0.5594	0.5599	0.5652
MAE Conv	0.5494	0.5598	0.5498	0.5467	0.5464	0.5496	0.5412	0.5423	0.5442	0.5417
MAE LSTM	0.516	0.5204	0.5198	0.5155	0.5153	0.5197	0.5159	0.5211	0.5103	0.5199

**Table 9 sensors-23-09585-t009:** Summary of the performance metrics MAE, MSE, and RMSE among the different NN (Linear, Dense, Conv. and LSTM).

	MAE	MSE	RMSE
Linear	0.6831	1.0354	1.0175
Dense	0.6661	1.0208	1.0104
Conv	0.635	0.9439	0.9716
LSTM	0.6214	0.9053	0.9515

## Data Availability

http://www.uv.es/eco4rupa/dataset.html, accessed on 26 October 2023. Please feel free to contact to the authors for further information.

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
