# Peer review of "AI-IoT Low-Cost Pollution-Monitoring Sensor Network to Assist Citizens with Respiratory Problems"

_sensors, 2023, doi:10.3390/s23239585_

Round 1

Reviewer 1 Report

Comments and Suggestions for Authors

The paper uses low-cost sensors to present an AI-IoT system for real-time air quality (AQ) monitoring. The goal is to increase the spatial density of measurements to provide localized AQ data and early warnings to citizens, especially those with respiratory issues. The system uses an MQTT-based architecture with ESP32 nodes connected to ZPHS01B AQ sensor modules. These nodes communicate via 5G technologies like NB-IoT and LTE-M to an ingestion server and database. To improve the accuracy of the raw sensor data, the system trains convolutional and deep neural networks (CNN, DNN) on both the raw data and measurements from official monitoring stations. The trained model provides 24-hour forecasts to enable proactive warnings about pollution exposure.

The introduction provides good motivation by citing statistics about the health impacts of poor AQ and the limitations of existing monitoring networks. Table 1 gives helpful background on major air pollutants and exposure limits. The authors make a compelling case for the need for higher-density AQ monitoring. The overall system architecture in Figure 7 is clear, as are the sample deployments next to reference stations in Figure 9. The paper gives practical technical implementation details on the MQTT protocol, security, database, and visualization. Figures 8 and 10 provide helpful snapshots of the live data collection. The authors evaluate four neural network architectures well. The LSTM network performed best, with average errors around 7.2%. The forecast result figures effectively demonstrate the system's capabilities.

So, the system design and concept are novel and well executed. The work addresses a significant problem and provides a practicable solution. The writing is clear and well-structured. The background and related work sections show good knowledge of the state-of-the-art. The figures and results support the claims appropriately.  After reading this paper, I have the following comments:

Major Comments:

1. More details are needed on the specific sensors used. For example, the sensor module ZPHS01B contains individual sensors for each pollutant. Knowing the sensor types, measurement principles, models, manufacturers, and performance characteristics would be helpful. This would help readers better evaluate the capabilities and limitations of the system.

2. The pre-processing, training methodology, and model optimization process for the neural networks need more explanation. Details are needed on data splitting, input preparation, training times, validation schemes, hyperparameter tuning, regularization, etc. This is important for reproducing the work.

3. More quantitative performance metrics would strengthen the results beyond the MAE. Other metrics like RMSE, R-squared, feature importance, confusion matrices, etc., would help better characterize the model's accuracy. Statistical significance testing should also be included.

4. More discussion about real-world deployment considerations is needed - sensor calibration requirements, maintenance needs, environmental robustness, power consumption, costs, etc. Some practical deployment challenges and mitigation strategies should be addressed.

5. The introduction highlights assisting citizens with respiratory issues as a critical goal. However, it is unclear if/how the system provides direct value to citizens regarding apps, alerts, health recommendations, etc. This aspect needs further development.

6. The choice of 24-hour forecasts should be justified in more detail. Could shorter or longer-term forecasts also be helpful? Why was 24 hours chosen as the most appropriate?

7. More details are needed about the broader IoT platform used. For example, what specific database, visualization, and other tools were utilized? This would aid reproducibility.

Minor Comments:

1.     Expand acronyms on first use, e.g., AQ, IoT, RNN.

2.     In some cases, figure quality could be improved with higher resolution and more descriptive captions. 

3.     Some grammar issues exist, and it is recommended to thoroughly proofread/edit for typos, redundancies, and clarity.

4.     References are comprehensive, but the style is inconsistent. Suggest conforming to journal guidelines. 

5.     The Conclusions section could provide more specific future work directions beyond improving the dataset and sensors.

Comments on the Quality of English Language

Moderate editing of the English language is required.

Reviewer 2 Report

Comments and Suggestions for Authors

More results are needed in the abstract ? 

Lines 31 -32 : Authors need to add reference regarding the percentage in question and not only a reference regarding the WHO guidelines

Lines 32- 34 : Sentence should be modified

Table 1 : Particles (PM) can also have a size less than 2.5 um. The sentence should be modified

Table 1 : The information presented in this table can be moved to the supplementary or written in brief in the text. What is presented is not used in the paper. 

Liness 41 -42 : Heavy metals are part of the particulate matter

Line 43 : The information presented by the authors is speculative. Although fossil fuel combustion is an important source of emission in urban areas, it is not the only one. Authors should search for references in order to quantify the contribution of fossil fuel combustion to the total PM concentration or the gas phase. 

The figure of the air quality stations can also be moved to the supplementary information

The pros and cons od the usage of low-cost AQ sensors can be put in the manuscript. it is not necessary to have a table for that

Line 103 : It is better to mention the pollutants again 

Table 3 : PM cannot be considered as a gas! The title of the second column of this table should be modified

Line 116 : PM is not a gas. It is the particulate phase of atmospheric pollutants

Line 141 : It is important to add maybe some highlights of the recommendations found in the reference in order to understand which type of recommendation it is

Figures 5 and 6 can be moved to the supplementary information

Figure 8 : The data presented in the screenshot should be explained. What are these values? and what the corresponding units of each parameter? 

Line 300 : Did you compare the data obtained by the AQ low cost sensors and the data from the official AQ monitoring stations?

Figure 10 : It is important to add the units of the different parameters

Figures 14-17: How can we explain that sometimes the predicted paramter value is different than the measured one? IS it because of a specific event happening at the sampling site? 

Comments on the Quality of English Language

Should be improved

Reviewer 3 Report

Comments and Suggestions for Authors

This study discusses the potential of low-cost air quality (AQ) sensors in wireless sensor networks (WSNs). However, these sensors suffer from complex data collection and analysis problems for AQ monitoring. With proper processing, they can provide valuable data. The goal of the study was to predict AQ readings for the next 24 hours with the help of a low-cost sensor network, combined with 5G communication and artificial intelligence techniques (using convolutional and deep neural networks, CNNs and DNNs) to warn residents in dangerous areas in advance. The study used a large-scale dataset for training and testing and achieved good results. This paper emphasizes the importance of AQ monitoring and its efficiency and accuracy through the application of low-cost sensors and AI technologies. The comments are below:

1, The abstract part of the paper has introduced the topic and objectives of the study, but the description of the innovations needs to be more specific, such as how the techniques are used and their uniqueness among the existing methods.

2,Some artificial intelligence techniques are used in the text, please give more algorithm-related content, algorithm-related formulas, algorithm complexity and so on.

3, The description of the algorithm is not clear in the text.

4, the description of the prediction result curve is missing.

5, A lot of background information is given in the beginning of the introduction, which is suggested to be curtailed.

6, Table 1 gives some very standard things that can be easily retrieved from the open literature. It is recommended to delete or abbreviate it if it is not absolutely necessary.

Comments on the Quality of English Language

It is recommended to check the English structure of the whole manuscript.

Round 2

Reviewer 1 Report

Comments and Suggestions for Authors

Good job.

Comments on the Quality of English Language

Moderate editing of the English language required.

Reviewer 2 Report

Comments and Suggestions for Authors

The authors have adressed the different comments ! Nothing to add 

Comments on the Quality of English Language

-

Reviewer 3 Report

Comments and Suggestions for Authors

The authors have improved the manuscript with more clarifications.

Comments on the Quality of English Language

Minor editing of English language required